# Evaluating the long-term consequences of air pollution in early life: geographical correlations between coal consumption in 1951/1952 and current mortality in England and Wales

David I W Phillips,[1] Clive Osmond,[1] Humphrey Southall,[2] Paula Aucott,[2] Alexander Jones,[3] Stephen T Holgate[4]

[1]The Medical Research Council's Lifecourse Epidemiology Unit, University of Southampton, Southampton, UK
[2]Department of Geography, University of Portsmouth, Portsmouth, UK
[3]Department of Paediatrics, University of Oxford, Oxford, UK
[4]Department of Medicine, University of Southampton, Southampton, UK

**Correspondence to**
Professor David I W Phillips; diwp@mrc.soton.ac.uk

## ABSTRACT

**Objective** To evaluate associations between early life air pollution and subsequent mortality.

**Design** Geographical study.

**Setting** Local government districts within England and Wales.

**Exposure** Routinely collected geographical data on the use of coal and related solid fuels in 1951–1952 were used as an index of air pollution.

**Main outcome measures** We evaluated the relationship between these data and both all-cause and disease-specific mortality among men and women aged 35–74 years in local government districts between 1993 and 2012.

**Results** Domestic (household) coal consumption had the most powerful associations with mortality. There were strong correlations between domestic coal use and all-cause mortality (relative risk per SD increase in fuel use 1.124, 95% CI 1.123 to 1.126), and respiratory (1.238, 95% CI 1.234 to 1.242), cardiovascular (1.138, 95% CI 1.136 to 1.140) and cancer mortality (1.073, 95% CI 1.071 to 1.075). These effects persisted after adjustment for socioeconomic indicators in 1951, current socioeconomic indicators and current pollution levels.

**Conclusion** Coal was the major cause of pollution in the UK until the Clean Air Act of 1956 led to a rapid decline in consumption. These data suggest that coal-based pollution, experienced over 60 years ago in early life, affects human health now by increasing mortality from a wide variety of diseases.

## INTRODUCTION

Although air pollution has detrimental effects on health at all ages,[1] there is accumulating evidence that the consequences of early life exposure may be more important than that occurring at other stages in life. Developing tissues and organs are particularly susceptible due to their high cell proliferation rates in prenatal life and infancy. Postnatally, infants and young children have increased vulnerability due to immature detoxification systems,

### Strengths and limitations of this study

► The first to evaluate the very long-term (60 years) mortality resulting from air pollution experienced in early life.
► Use of national mortality data with virtually complete ascertainment and a large number of deaths (>3.5 million).
► Coal consumption data provide an integral of air pollution over a 12-month period.
► The analysis also allows for current pollution levels.
► The study depends on the use of geographical correlations together with census-derived measures of socioeconomic status; individual-level data are not available.

high infection rates and patterns of behaviour that increase their pollutant exposure.[2] Pregnant mothers exposed to air pollution are more likely to give birth prematurely and have offspring with intrauterine growth retardation and poor growth in infancy.[3 4] Such outcomes are linked with long-term adverse health effects, including increased risks of respiratory and cardiovascular disease.[5] An increasing body of evidence from both the medical and economic literature provide support for the adverse effects of early life pollutant exposure on a variety of health and human capital outcomes in later life.[2 6] This has important public health and policy implications in the high-income world and for resource-poor countries where open fires and the consequent exposure of mothers and young children to pollutants is widespread.[2 7 8]

In common with many other countries, the UK had high levels of air pollution in the past.[9] Most of this was due to coal use in industry and as a domestic fuel, which gave rise to the 'great stinking fogs' that previously

characterised London and other major cities.[9] The London smog of 1952 killed an estimated 12 000 people[10] and led to the first UK Clean Air Act being passed in 1956. Following these events, a national survey of smoke and $SO_2$ concentrations was set up during the 1960s to monitor the progress of pollution control following the Act. From around 1960, there was a progressive reduction in black smoke and $SO_2$ emissions, with nitrogen oxides, ozone and microparticulates from transport becoming the dominant forms of air pollution.[11]

By the middle of the 20th century, >200 million tonnes of coal were being used per year in England and Wales[12] and yet little is known about the long-term health effects of the combustion of such vast quantities of solid fuel. Studies published in the 1950s and 1960s suggested that there were strong cross-sectional correlations of the amounts of coal used in domestic fires with both childhood respiratory disease[13] and many major causes of adult mortality,[14 15] but it is not known if these effects have persisted. The present study is based on a dataset compiled by the Ministry of Fuel and Power that recorded the quantities of different types of solid fuel burnt annually between May 1951 and May 1952 in 1145 areas of England and Wales. The extent to which exposure to the resulting pollutants may have influenced present day health was assessed by relating these data to mortality rates in the same areas 50–60 years later.

## METHODS

Shortages of coal following World War II led to the continuation of a system of rationing and distribution that had been initiated in 1939. This system was controlled by local fuel overseers (LFOs) who were appointed for each local government district (LGD) or group of districts. The Ministry of Fuel and Power published estimates of the quantities of different types of solid fuel burnt annually in each of the 1145 LFO areas between May 1951 and May 1952 in England and Wales—data were only published for these years.[16] The tables list the consumption of solid fuels classified by the main consumer groups in each LFO area. Domestic (household) supplies included coal, the amounts used in each LFO area based on the number of registered premises, and other solid fuels including coke, briquettes, ovoids and similar patent fuels where consumption was based on merchant's disposals. The annual household coal allowance was higher in the North (2540 kg) than the South (1727 kg). Miners' concessionary coal was also included in the estimates. Industrial fuels were predominantly coal and coke and there were separate listings of the quantities of solid fuel used in electricity generation and for carbonisation or briquetting plants. The amounts of smokeless fuel (anthracite, dry steam coal, coke, Phurnacite, Coalite and Rexco) used in domestic and industrial premises were recorded separately.

The Office for National Statistics (ONS) provided extracts from all death certificates in England and Wales

for each of the current LGDs during 1993–2012. This is the period covered by the ninth and tenth revisions of the International Classification of Disease (ICD). The ICD codes used to define causes of death are mentioned in online supplementary appendix 1. Mortality rates were calculated using population estimates provided by the ONS for each LGD. The main technical problem was to relate the historical 1145 LFO areas to the current 342 LGDs, following successive local government reorganisations in 1974, the 1990s and 2009, together with numerous individual boundary changes. Detailed digital boundaries for the LGDs of England and Wales used in the 1951 census have already been constructed as part of the Great Britain Historical Geographical Information System (GIS).[17 18] As the LFO areas either exactly corresponded to LGDs or were defined as aggregates of them, constructing digital boundary data for LFO areas was straightforward. However, modern LGDs are not necessarily simple aggregates of the LFO areas, so the first stage of the redistricting procedure was to reallocate the fuel consumption data to the 12 504 Civil Parishes of 1951. Digital boundaries already existed for the parishes, and the reallocation was based on what proportion each parish's total population, as listed by the 1951 census, was of the overall population of the containing LFO area. The fuel data were then reallocated to the modern areas by digitally overlaying the two sets of boundaries: whatever proportion of a 1951 parish's area fell within a modern district, that proportion of the parish's fuel consumption was assigned to the district. Air pollution from each source was estimated by calculating the consumption of solid fuel per acre.

Historically, air pollution was worse in predominantly industrial towns with a high population density; these towns also tended to have large populations of factory workers who left school early and worked in low status occupations.[14] These are important potential confounders in a study of air pollution and disease, as poverty and overcrowding are known to be associated with many of the major causes of mortality. To control for these, we used a series of variables extracted from the published reports of the 1951 census: male unemployment; indices of social class; overcrowding, assessed as the proportion of households with more than one person per room; density, the number of people per acre and educational achievement, indicated by the proportion of employed men in each area who left school before the age of 15 years. Social class was based on the Registrar General's occupation-based system that categorises people into five ordinal groups: (i) professional occupations, (ii) managerial and technical occupations, (iii) skilled occupations, (iv) partly skilled occupations and (v) unskilled occupations which were combined into a weighted score.[19] The census data were redistricted in the same way as for the fuel. Parallel data were obtained from the 2001 census: unemployment rate, social grade (combined into a weighted score), proportion of households with >1 person/room, persons/hectare and proportion with no qualifications in

each of the 342 LGDs. We also allowed for the impact of current levels of air pollution, which were obtained from published data reporting the modelled annual average concentrations of fine particulate matter ($PM_{2.5}$) in each of the 342 LGDs.[20]

## Statistical methods

We studied deaths in people aged between 35 and 74 years, as this was the approximate age range of the generation who were in their first decade of life in 1951–1952, and excluded older age groups where the cause of death was less likely to have been ascertained accurately.[21] We calculated the number of deaths expected for each 5-year age and sex group within the LGDs by multiplying the number of people in each group by the national age-specific and sex-specific death rates in 1993–2012. We expressed the number of observed deaths in each area as a percentage of the expected deaths for that group, that is, as standardised mortality ratios (SMRs). We used Poisson regression to model SMRs, expressing the effect of fuels and other variables on mortality in terms of the hazard ratio (or relative risk) per SD change in the explanatory variable. This weights the analyses appropriately to allow for variation in population size across LGDs. A Fisher-Yates transform[22] was used to standardise pollution and socioeconomic variables. This yielded variables with zero means, unit SDs and symmetrical distributions, allowing direct comparison of the strength of associations with these variables.

## Patient and public involvement

No patients or public were involved in this study.

## RESULTS

There were a total of 3 535 136 deaths in England and Wales during the 20-year period from 1993 to 2012 in the age group of 35–74 years. Mortality rates (SMR) in the 342 LGDs ranged from 64.7 in East Dorset to 153.1 in Manchester. Table 1 shows the different uses of solid fuel during 1951/1952 in England and Wales. Of the over 200 million tonnes distributed, 37.6% was used in industrial processes, 16.6% in electricity generation and 27.5%

in carbonisation or briquetting processes. Only 18.3% was used domestically. Smokeless fuel accounted for approximately 32.1% of industrial and 14.3% of domestic consumption. Figure 1 is a map of England and Wales showing the quintiles of domestic fuel consumed per acre. This was generally low in rural areas but was greatest in London and in the former industrial areas of the UK (South Wales, the Midlands, Lancashire, the West Riding of Yorkshire and Tyneside). The mean current $PM_{2.5}$ concentrations in the 342 areas was 9.48 (SD 1.67) $\mu g/m^3$ and ranged from 5.5 in Gwynedd (Wales) to 14.9 $\mu g/m^3$ in the City of Westminster, London.

Table 2 shows the relationship between the various sources of fuel burnt in 1951/1952 (excluding smokeless fuels) and subsequent all-cause mortality or mortality due to respiratory, cardiovascular disease and cancer in the 342 areas. The influence of the 1951 census socioeconomic indicators (social class, education, crowding, density and unemployment) are also shown. In univariate analyses, all variables were significantly associated with both all-cause and cause-specific mortality rates, although the effects of domestic and industrial consumption tended to be most strongly related to the mortality outcomes. Multivariate analyses were carried out to evaluate the relative effects of fuel usage and the major confounding variables derived from the 1951 census. This analysis shows that domestic consumption was strongly and significantly associated with both all-cause and cause-specific mortality rates, an effect which was markedly stronger than the other sources of fuel. Smokeless fuels represented <15% of the total fuels used (table 1) and in separate regression analyses had no statistically significant independent effect on mortality rates (data not shown).

There was a very high correlation (r=0.97) between domestic consumption and population density, resulting in unacceptable collinearity in regression analyses that contained both variables. This occurred because both had the same denominator, the area of the LGD. To allow the influence of both of these factors to be modelled statistically, we created two further variables: the sum of and the difference between domestic use and density. These variables were uncorrelated (r=0.0). These were entered

| Table 1 | Fuel consumption in England and Wales 1951/1952 | | |
|---|---|---|---|
| | **Total annual consumption (tonnes×1000)** | **Annual consumption per acre in the 342 LGDs (tonnes)** | |
| | | **Median** | **Quartiles** |
| Total domestic | 36 783 | 1.00 | 0.29, 4.38 |
| (Smokeless) | 5247 | 0.17 | 0.05, 0.63 |
| Total industrial | 75 597 | 1.17 | 0.23, 5.58 |
| (Smokeless) | 24 294 | 0.47 | 0.12, 2.07 |
| Carbonisation and briquetting | 55 122 | 0.63 | 0.10, 5.36 |
| Electricity generation | 33 304 | 0.07 | 0.00, 1.85 |
| Total | 200 806 | 4.21 | 0.82, 21.38 |

LGD, local government district.

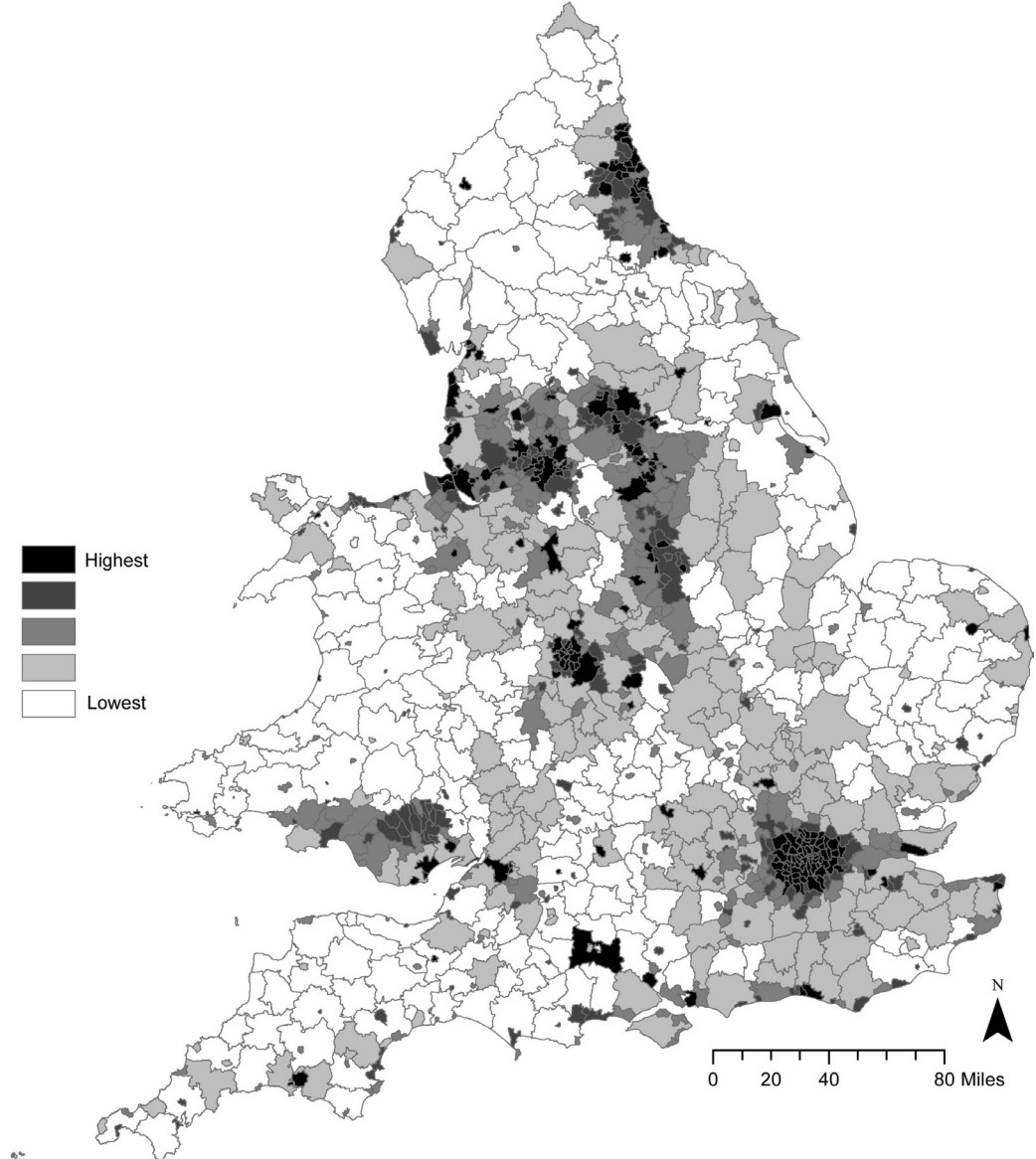

**Figure 1** Fifths of domestic fuel consumption in England and Wales, 1951–1952.

into the regression models in table 2, which shows both the unadjusted coefficients and the coefficients adjusted for all other variables (table 2, lower panel). The positive regression coefficients for the sum variable show that the joint effects of domestic use and density on all four causes of mortality were powerful. However, associations with the difference variable suggest that the influence of domestic use was stronger than that of density.

Table 3 shows the relationship between current socioeconomic indicators or microparticulate air pollution and all-cause or cause-specific mortality. The lower section of table 3 shows that the effect of domestic coal usage is attenuated by successive adjustments for socioeconomic indicators from the 1951 census, for indicators from the 2001 census together with particulate exposure and for both of these sets of factors. However, in all cases the association with domestic coal consumption remained strong and statistically significant.

The effects of the potential confounders were tested singly and in combination. There was no evidence of non-linear relationships. Figure 2 shows the associations between quartiles of domestic fuel consumption and all-cause mortality or the major mortality categories. These associations were progressive and remained strong and statistically significant for each cause group after adjustment for social class, education, crowding, unemployment in 1951, and for current socioeconomic indicators and $PM_{2.5}$ concentrations. They were present in both genders and were similar during the ICD9 and ICD10 periods.

Tables 4 and 5 show the relationship between domestic fuel consumption and specific causes of mortality. As before, the data are shown before and after adjustment for past and current socioeconomic indicators and current $PM_{2.5}$ levels. The respiratory conditions responsible for most deaths, chronic obstructive pulmonary

**Table 2** Association between fuel consumed in 1951/1952 or socioeconomic indicators derived from the 1951 census and current causes of mortality in the 342 LGDs in England and Wales

| | Relative risk of mortality per SD change in fuel usage or socioeconomic indicator (95% CI) | | | |
| --- | --- | --- | --- | --- |
| | All cause | Respiratory (ICD9: 460–519, ICD10: J00–J99) | Cardiovascular (ICD9: 390–459 ICD10: I00–I99) | Cancer (ICD9: 140–208, ICD10: C00–C97) |
| **Univariate analyses** | | | | |
| **Fuel consumed 1951/1952** | | | | |
| Domestic (ex. smokeless) | 1.124 (1.123 to 1.126) | 1.238 (1.234 to 1.242) | 1.138 (1.136 to 1.140) | 1.073 (1.071 to 1.075) |
| Industrial (ex. smokeless) | 1.129 (1.128 to 1.131) | 1.244 (1.239 to 1.248) | 1.152 (1.150 to 1.154) | 1.080 (1.079 to 1.082) |
| Carbonisation | 1.120 (1.119 to 1.122) | 1.225 (1.221 to 1.230) | 1.135 (1.133 to 1.137) | 1.073 (1.071 to 1.074) |
| Electricity generation | 1.096 (1.095 to 1.097) | 1.181 (1.177 to 1.185) | 1.110 (1.107 to 1.112) | 1.054 (1.052 to 1.056) |
| **1951 census** | | | | |
| Social class score* | 1.117 (1.115 to 1.118) | 1.190 (1.186 to 1.194) | 1.144 (1.142 to 1.146) | 1.087 (1.085 to 1.089) |
| Low educational level | 1.093 (1.092 to 1.094) | 1.154 (1.150 to 1.158) | 1.125 (1.123 to 1.127) | 1.064 (1.062 to 1.066) |
| Crowding | 1.106 (1.105 to 1.108) | 1.179 (1.175 to 1.183) | 1.126 (1.124 to 1.128) | 1.075 (1.073 to 1.077) |
| Density | 1.105 (1.104 to 1.106) | 1.206 (1.202 to 1.211) | 1.113 (1.111 to 1.115) | 1.059 (1.057 to 1.061) |
| Unemployment | 1.074 (1.073 to 1.076) | 1.113 (1.109 to 1.117) | 1.077 (1.075 to 1.079) | 1.055 (1.053 to 1.057) |
| **Multivariate analysis** | | | | |
| **Fuel consumed 1951/1952** | | | | |
| Domestic (ex. smokeless) | 1.061 (1.058 to 1.064) | 1.127 (1.118 to 1.137) | 1.051 (1.046 to 1.056) | 1.027 (1.023 to 1.031) |
| Industrial (ex. smokeless) | 1.008 (1.006 to 1.011) | 1.031 (1.023 to 1.039) | 1.020 (1.016 to 1.025) | 1.011 (1.007 to 1.015) |
| Carbonisation | 1.029 (1.027 to 1.030) | 1.044 (1.038 to 1.050) | 1.029 (1.026 to 1.033) | 1.019 (1.016 to 1.022) |
| Electricity generation | 1.002 (1.000 to 1.003) | 0.999 (0.994 to 1.004) | 1.005 (1.002 to 1.008) | 0.995 (0.992 to 0.997) |
| **1951 census** | | | | |
| Social class | 1.062 (1.060 to 1.064) | 1.094 (1.087 to 1.101) | 1.058 (1.054 to 1.062) | 1.054 (1.051 to 1.058) |
| Education | 1.021 (1.019 to 1.022) | 1.027 (1.021 to 1.032) | 1.045 (1.042 to 1.048) | 1.007 (1.005 to 1.010) |
| Crowding | 1.005 (1.004 to 1.007) | 1.001 (0.996 to 1.006) | 1.012 (1.009 to 1.014) | 1.007 (1.005 to 1.010) |
| Unemployment | 1.032 (1.031 to 1.034) | 1.037 (1.032 to 1.042) | 1.036 (1.034 to 1.039) | 1.027 (1.025 to 1.030) |
| **Combined effects analysis of domestic usage and density** | | | | |
| **Unadjusted** | | | | |
| Sum | 1.112 (1.111 to 1.113) | 1.219 (1.215 to 1.223) | 1.122 (1.120 to 1.124) | 1.064 (1.062 to 1.066) |
| Difference (domestic–density) | 1.074 (1.073 to 1.076) | 1.117 (1.113 to 1.121) | 1.095 (1.093 to 1.097) | 1.052 (1.050 to 1.054) |
| **Adjusted†** | | | | |
| Sum | 1.054 (1.051 to 1.056) | 1.116 (1.107 to 1.126) | 1.042 (1.037 to 1.047) | 1.022 (1.017 to 1.026) |
| Difference (domestic–density) | 1.031 (1.029 to 1.032) | 1.048 (1.044 to 1.053) | 1.036 (1.034 to 1.038) | 1.018 (1.016 to 1.020) |

*See 'Methods' section.
†Adjusted for social class, education, crowding, unemployment, industrial usage and fuel used in carbonisation and electricity generation.
ICD, International Classification of Disease; LGD, local government district.

disease (COPD) and pneumonia were strongly and consistently associated with domestic fuel consumption while associations with asthma mortality were somewhat weaker. However, the largest relative risks were with tuberculosis.

While all four major causes of cardiovascular mortality were significantly associated with domestic consumption, the strongest relationship was with rheumatic heart disease.

**Table 3** Association between socioeconomic indicators derived from the 2001 census or current $PM_{2.5}$ concentrations and current causes of mortality in the 342 LGDs in England and Wales together with multivariate models evaluating the association between domestic coal usage and mortality following adjustment for the 1951 and 2001 socioeconomic indicators and current $PM_{2.5}$ exposure

| | Relative risk of mortality per SD change in socioeconomic indicator, particulate or fuel usage (95% CI) | | | |
|---|---|---|---|---|
| | **All cause** | **Respiratory (ICD9: 460–519, ICD10: J00–J99)** | **Cardiovascular (ICD9: 390–459 ICD10: I00–I99)** | **Cancer (ICD9: 140–208, ICD10: C00–C97)** |
| Univariate analyses | | | | |
| 2001 census | | | | |
| Social class score* | 1.139 (1.137 to 1.140) | 1.221 (1.217 to 1.226) | 1.170 (1.168 to 1.173) | 1.098 (1.096 to 1.100) |
| Low educational level | 1.122 (1.121 to 1.123) | 1.195 (1.191 to 1.199) | 1.157 (1.155 to 1.159) | 1.087 (1.085 to 1.089) |
| Crowding | 1.101 (1.099 to 1.102) | 1.196 (1.192 to 1.201) | 1.117 (1.115 to 1.119) | 1.052 (1.051 to 1.054) |
| Density | 1.101 (1.100 to 1.102) | 1.196 (1.192 to 1.200) | 1.107 (1.104 to 1.109) | 1.059 (1.057 to 1.061) |
| Unemployment | 1.157 (1.156 to 1.158) | 1.270 (1.266 to 1.275) | 1.184 (1.182 to 1.186) | 1.099 (1.097 to 1.101) |
| Current $PM_{2.5}$ | 1.027 (1.026 to 1.028) | 1.076 (1.073 to 1.080) | 1.028 (1.027 to 1.030) | 1.005 (1.003 to 1.006) |
| Multivariate analyses | | | | |
| Model 1 (effect of domestic usage allowing for 1951 socioeconomic indicators) | 1.094 (1.092 to 1.095) | 1.195 (1.190 to 1.200) | 1.097 (1.095 to 1.100) | 1.038 (1.033 to 1.043) |
| Model 2 (effect of domestic usage allowing for 2001 socioeconomic indicators and current $PM_{2.5}$) | 1.081 (1.078 to 1.084) | 1.179 (1.167 to 1.190) | 1.088 (1.082 to 1.093) | 1.038 (1.033 to 1.043) |
| Model 3 (effect of domestic usage allowing for 1951 and 2001 socioeconomic indicators and current $PM_{2.5}$) | 1.084 (1.080 to 1.087) | 1.189 (1.177 to 1.201) | 1.088 (1.083 to 1.094) | 1.037 (1.032 to 1.042) |

*See 'Methods' section.
ICD, International Classification of Disease; LGD, local government district; PM, particulate matter.

The data on associations of domestic fuel use with mortality from specific cancers were striking (table 5). There were strong relationships with epithelial cancers; lip, oral cavity and pharynx (adjusted HR, 1.207); laryngeal cancers (1.263) and lung (1.126). The upper gastrointestinal cancers, oesophagus (1.088), stomach (1.117) and liver (1.053), were more strongly associated with domestic pollution than the lower gastrointestinal cancers. In contrast, the major urinary, reproductive and haematological cancers showed no increase in risk, with the exception of cervical cancer (1.131). Some cancers, for example, melanoma and brain, were associated with a somewhat reduced relative risk.

## DISCUSSION
Areas of the UK that had high domestic consumption of coal in 1951/1952 now have raised mortality from a wide variety of causes, including cardiovascular and respiratory diseases and certain cancers. The correlations are strong, statistically significant and independent of all available variables that might be considered major confounders, whether assessed in 1951 or currently, including social class, level of education, overcrowding and unemployment.

Very few comparable data exist in the literature, particularly with such a long follow-up period. Our findings accord with an increasing body of evidence that early life exposure to air pollution has detrimental long-term health effects.[6 15 23] In particular, two recent studies demonstrated important findings. In the first, children exposed to the great London Smog of 1952 were found to have greater risk of asthma in adulthood, compared with unexposed children.[24] In the second, which was based on the ONS longitudinal study, air pollution concentrations

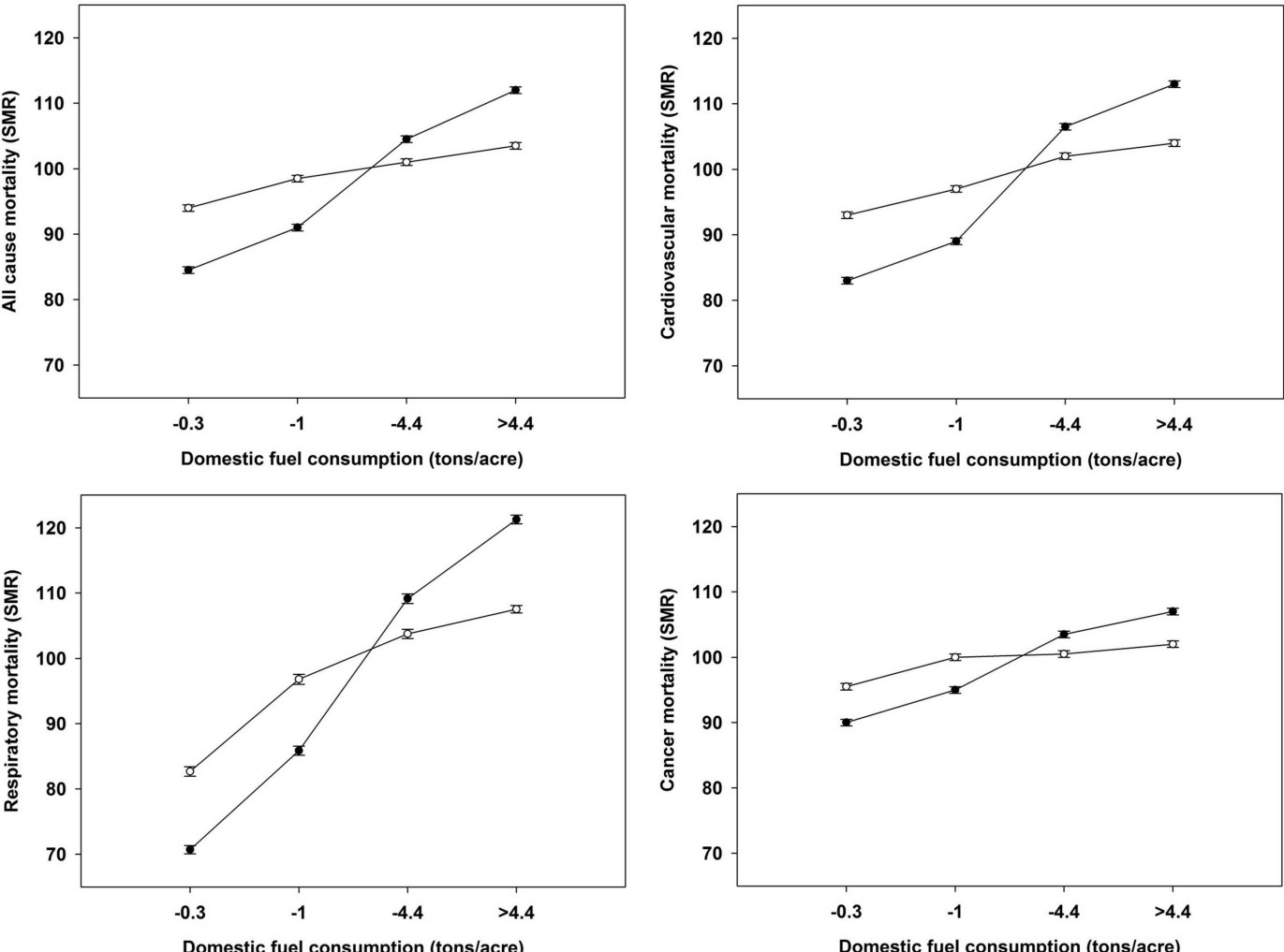

**Figure 2** Association between quartiles of domestic fuel consumption (ex smokeless) and both all-cause and the major causes of mortality in men and women aged 35–74 years in the 342 local government districts. (●unadjusted, ○adjusted for socioeconomic indicators in 1951 and 2001 and current $PM_{2.5}$; errors are 95% CI). SMR, standardised mortality ratio.

at the site of residence, assessed every 10 years from 1971, were associated with increased total, respiratory and cardiovascular mortality.[25] Our conclusions also accord with historical studies of coal use and mortality in Britain, which estimated that each SD increase in coal use raised mortality by 5%–15% in infants and 5% in adults.[26]

Our study necessarily depended on published crude estimates of coal consumption as an index of domestic air pollution. We contend that it is very reasonable to assume that the quantity of coal burnt relates to the amount of pollution emitted, although local factors such as climate and wind strength would also have a major influence. The consumption measures in our study were shown to correlate with subsequent measurements of pollution in the UK in the early 1960s,[13] when coal was still a major fuel and source of pollution. This approach is also supported by a number of studies carried out over the last 60 years, showing correlations with diverse health outcomes including reduced early growth,[3] chronic bronchitis and lung cancer.[13–15] While the consumption data come from a short time-period between 1951 and 1952,

they are likely to reflect smoke emission over many years. The UK coal production was fairly constant during the late 1940s, declined a little during the 1950s and only fell dramatically during the 1960s and subsequent decades.[12] In the postwar years, economic necessity drove the use of low-grade, bituminous domestic coal while better-quality 'hard' coals were exported. A typical, inefficient domestic grate burning low-grade coal would have produced lots of smoke, rich in a wide variety of potentially toxic compounds, including heavy metals, sulfur and complex mixtures of aliphatic and aromatic hydrocarbons.[27] Confirmation of this can be found in the lung tissue from autopsies of people exposed to the 1952 London smog, which was shown to contain both ultrafine carbonaceous and metal PM.[28] Children would have been exposed both in the home and in their local environment, although it is not clear from our data which of these would have been the most important source. However, our study suggests that it was domestic fuel consumption that had the greatest long-term adverse health effects, with industrial pollution or pollution from power stations or other

**Table 4** Association between domestic fuel usage and major causes of non-cancer mortality among men and women aged 35–74 years in the 342 areas

| | Relative risk per SD increase in domestic usage | | | |
|---|---|---|---|---|
| | Unadjusted (95% CI) | | Adjusted* (95% CI) | |
| **Respiratory** | | | | |
| Chronic obstructive pulmonary disease | 1.253 | 1.245 to 1.260 | 1.142 | 1.122 to 1.163 |
| Asthma | 1.199 | 1.176 to 1.223 | 1.093 | 1.032 to 1.158 |
| Pneumonia | 1.262 | 1.254 to 1.269 | 1.227 | 1.205 to 1.250 |
| Tuberculosis | 1.734 | 1.676 to 1.793 | 1.298 | 1.167 to 1.444 |
| **Cardiovascular** | | | | |
| Ischaemic heart disease | 1.141 | 1.138 to 1.143 | 1.102 | 1.095 to 1.110 |
| Rheumatic heart disease | 1.209 | 1.184 to 1.234 | 1.240 | 1.166 to 1.318 |
| Stroke | 1.137 | 1.133 to 1.141 | 1.114 | 1.101 to 1.127 |
| Hypertension | 1.204 | 1.196 to 1.212 | 1.112 | 1.089 to 1.135 |

*Adjusted for 1951 and 2001 socioeconomic indicators and current $PM_{2.5}$ exposure.
PM, particulate matter.

sources causing relatively smaller effects (table 2). The probable explanation for this is that most industrial and power station pollution was vented through tall chimneys, decreasing ground-level pollution in approximate proportion to the inverse square of chimney height.[29]

Current levels of pollution in the 342 local areas were based on ambient levels of fine $PM_{2.5}$ because large cohort studies have shown that this measure correlates best with long-term health effects of pollution.[30–32] The effects of $PM_{2.5}$ on current mortality (table 3) in the present study are consistent with previously published estimates,[20] but importantly the mortality correlations with coal consumption persisted after adjustment for this measure of current pollution.

Our data do not allow determination of the age at which children were most vulnerable to the effects of pollution, although a cohort-based analysis (figure 3) suggests that exposures around the time of birth or infancy are likely to have been most important, especially for all-cause and respiratory mortality. We can also only speculate on the mechanism by which air pollution arising from coal consumption, predominantly in a domestic setting, might increase the risk of mortality. There may have been direct mutagenic effects, especially in relation to the increased cancer risk. Another possibility is that the pollutants triggered mechanisms that selected alternative developmental pathways in the young, perhaps through epigenetic modification of gene expression.[33]

Elevated relative risks were observed for respiratory diseases, especially COPD, asthma and pneumonia (table 4), which is consistent with past evidence that black smoke exposure is a risk factor for these conditions.[14 15 23] The strongest association we found, however, was with tuberculosis, which is less supported by existing literature. A recent systematic review and meta-analysis concluded that the evidence for such an association was inconclusive, but it was reliant on case-control and cross-sectional studies.[34] The findings in the present study raise the possibility that early exposure to air pollution could be a critical factor in determining susceptibility to tuberculosis, although other factors such as population density cannot be completely excluded. Cardiovascular disease was also found to be strongly associated with domestic fuel consumption, which accords with earlier UK findings,[15 23] the Harvard Six Cities study[31] and a multicentre study of European cohorts.[35] The strength of the association was similar with ischaemic heart disease and stroke. The association with rheumatic heart disease was strong and confirms the findings of a previous study based on a subset of large Country Boroughs in England and Wales, and further supports the suggestion that an increase in susceptibility to infectious diseases is linked with exposure to air pollution in infancy or early childhood.[36] An additional finding was a strong association with hypertension, which is supported by a meta-analysis that showed both short-term and long-term exposure to some air pollutants increases the risk of hypertension.[37] The data on cancers (table 5) show wide variation in the level of associated risk. They show that the strongest associations are observed with epithelial cancers of the respiratory system (lip, oral cavity and pharynx; larynx; trachea and lung) and the upper gastrointestinal system (stomach and liver). The major reproductive, urinary and haematological cancers were not positively associated with domestic pollution with the exception of cervical cancer, for which there is evidence that cigarette smoking is an important cofactor.[38] Malignant melanoma had a lower relative risk in the areas with high consumption possibly because of reduced sunlight exposure. Some cancers showed small reductions in relative risk, for example, brain cancer, and

**Table 5** Association between domestic fuel usage and cancer mortality in the 342 areas

| Site | Relative risk per SD increase in domestic usage | | | |
|---|---|---|---|---|
| | Unadjusted (95% CI) | | Adjusted* (95% CI) | |
| Lip, oral cavity and pharynx | 1.220 | 1.204 to 1.236 | 1.207 | 1.161 to 1.255 |
| Oesophagus | 1.038 | 1.030 to 1.046 | 1.088 | 1.064 to 1.112 |
| Stomach | 1.171 | 1.161 to 1.182 | 1.117 | 1.088 to 1.147 |
| Colon | 0.996 | 0.989 to 1.002 | 1.027 | 1.007 to 1.048 |
| Rectosigmoid junction, rectum and anus | 1.073 | 1.063 to 1.082 | 1.039 | 1.012 to 1.066 |
| Liver, gall bladder and bile ducts | 1.193 | 1.179 to 1.206 | 1.053 | 1.019 to 1.089 |
| Pancreas | 1.022 | 1.014 to 1.030 | 0.977 | 0.956 to 0.999 |
| Larynx | 1.347 | 1.318 to 1.376 | 1.264 | 1.185 to 1.348 |
| Trachea, bronchus and lung | 1.195 | 1.190 to 1.199 | 1.126 | 1.115 to 1.137 |
| Malignant melanoma | 0.885 | 0.873 to 0.898 | 0.889 | 0.854 to 0.924 |
| Breast | 0.990 | 0.984 to 0.995 | 0.958 | 0.943 to 0.974 |
| Cervix | 1.155 | 1.135 to 1.175 | 1.131 | 1.074 to 1.190 |
| Uterus | 1.032 | 1.015 to 1.049 | 0.930 | 0.886 to 0.975 |
| Ovary | 0.968 | 0.959 to 0.976 | 0.986 | 0.960 to 1.012 |
| Prostate | 0.989 | 0.981 to 0.998 | 0.962 | 0.939 to 0.986 |
| Kidney | 1.000 | 0.989 to 1.010 | 1.003 | 0.973 to 1.035 |
| Bladder | 1.056 | 1.044 to 1.068 | 1.024 | 0.991 to 1.057 |
| Brain, eye and central nervous system | 0.952 | 0.943 to 0.961 | 0.962 | 0.936 to 0.988 |
| Hodgkin's lymphoma | 1.030 | 0.994 to 1.067 | 0.919 | 0.829 to 1.018 |
| Non-Hodgkin's lymphoma | 1.024 | 1.014 to 1.034 | 0.952 | 0.925 to 0.979 |
| Multiple myeloma | 0.999 | 0.986 to 1.013 | 0.941 | 0.905 to 0.978 |
| Leukaemia | 1.008 | 0.998 to 1.019 | 0.961 | 0.931 to 0.991 |

*Adjusted for 1951 and 2001 socioeconomic indicators and current $PM_{2.5}$ exposure.
PM, particulate matter.

although the reasons are not clear, this may be due to residual confounding, which is evident because of the very large numbers of deaths and high sensitivity of this analysis.

There are a number of potential limitations to our study. Because neighbouring geographical areas tend to have similar levels of coal consumption, spatial autocorrelation may lead to CIs that are narrower than they should be. However, an analysis evaluating the influence of spatial autocorrelation (described in detail in online supplementary appendix 2) suggest that this is not a serious problem in the coal consumption data. Calculations incorporating the autocorrelation term into regression analyses increase the very narrow CIs, as shown in table 2, by 7.1%.

An inevitable issue with evaluating the association of early exposures with causes of late-life mortality is the long latency between, in this case, the measurements of coal consumption and the outcomes. It could be argued that internal migration might have weakened the associations that we observed as selective migration has been proposed as an explanation for geographical variations

in mortality. This is supported by some studies[39] but not others.[40 41] Migration is complex as, while migrants are generally healthier, those moving short distances tend to have higher mortality than those moving long distances.[42 43] Although we do not have migration data, most migration during the time period of this study occurred over short distances, with only a small minority migrating significant distances.[44] A more recent study of migration based on the 2001 census, while documenting the complex age and social class influences on migration, suggested that the declining industrial areas of South Wales, Yorkshire, Greater Manchester and Lancashire and the North-East of England, which were the areas that had high coal consumption, had in-migration and out-migration levels in all age groups that were far below the National average.[45] In addition, our analysis allows for the factors that tend to drive migration such as education and socioeconomic status.[46] Finally, it is difficult to see how selective migration could explain the specific links between coal consumption and some conditions and not others (table 5).

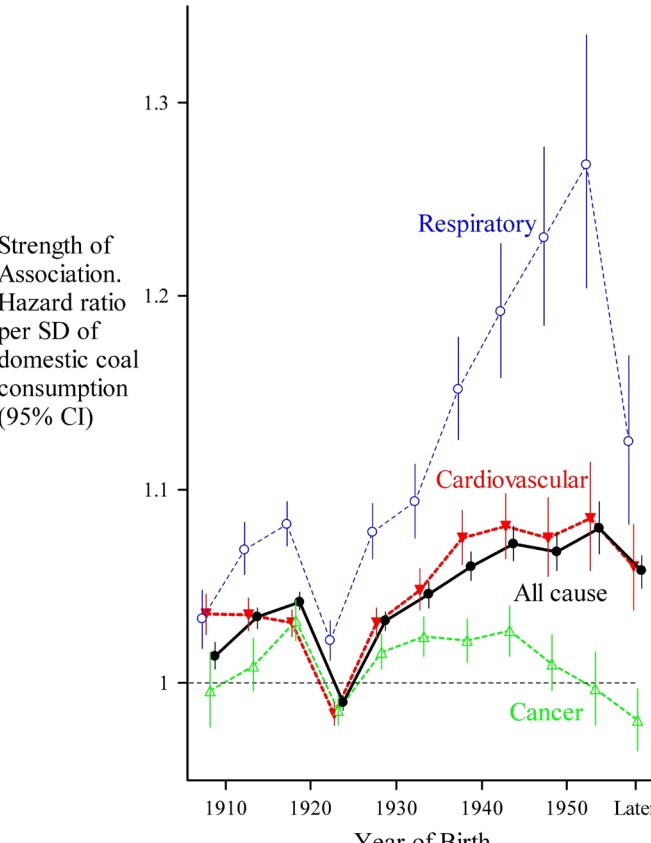

**Figure 3** The strength of the geographical correlations between domestic consumption and current mortality according to successive birth cohorts. For all-cause and respiratory mortality, the strongest associations were observed among those born in 1952/1953 suggesting that exposure around the time of birth had the greatest effect on mortality. For cardiovascular and cancer mortality, the peak associations were more blunted, suggesting that exposure over a wider age range was associated with subsequent mortality.

Any undetected bias in our observational study would be most likely due to confounding from an unknown source. However, a sensitivity analyses suggested that such a variable would have to correlate very strongly (r>0.5) with both domestic air pollution and respiratory disease, for example, to confound the relationships we have observed. There are no obvious candidates for such a confounder that were not addressed by our analysis, apart from air pollutants from sources other than the combustion of coal and solid fuels, measurements of which are unavailable for the early 1950s. As many of these would have been geographically localised and specific to certain industries, it is difficult to see how they could be confounders, especially as our analyses incorporate adjustment for socioeconomic factors (eg, occupation/urbanisation) that would reflect differential exposure to these substances. In addition, we have examined the effects of the major confounding variables that were available in the 1951 and 2001 census and that are likely to be related to both residence in an area of high pollution and to the various outcomes that we have

studied. The 1951 census contained several additional measures of household amenities, which were also examined in regression analyses. We found that these did not explain any further variance in our analyses beyond that due to the confounding variables that we used in the final analyses. Adjusting the risk estimates associated with domestic consumption for social class, density, unemployment and education reduced their magnitude, but they still remained strongly significant (table 3 and figure 2). However, it could be argued that some of these variables are so closely related to the measures of pollution that the regression-based adjustment has resulted in conservative risk estimates. Unsurprisingly, population density was highly correlated with domestic fuel use because fuel was rationed on a per household basis. In our analyses based on a published technique,[47] we were able to show that pollution had a more marked effect than population density but the close relationship between these variables prevented us from using this in the adjusted regression models and, as a result, we cannot confidently exclude the influence of early population density. However, we would also argue that, in this context, density (population per unit area) may be a less important confounder than overcrowding (proportion of households with >1 person per room). There is considerable observational evidence linking overcrowding with physical health, especially in adults[48] and in a review of the environmental factors most associated with health outcomes, Evans and Kantrowitz concluded that areal indices of density were less important than overcrowding in determining the health outcomes associated with high population density.[49] Finally, our data showing linear relationships between fuel usage and mortality (figure 2) are unlikely to be driven by the influence of density on pathogen transmission where saturation effects would be expected.[50] Although we did not have data on the prevalence of tobacco smoking, we do not think that this is likely to have seriously confounded our results. In the early 1950s, smoking was a social norm (in 1951 78% of men used tobacco), smoking was associated with low socioeconomic status,[51] which was included in our analyses and standardisation of the data for lung cancer rates as a proxy for smoking did not explain the correlations we observed.

## CONCLUSIONS

Although this is an ecological study based on geographical correlations, the results raise the possibility that domestic air pollution, experienced over 60 years ago by young children, affects human health now, by increasing mortality from a wide variety of diseases. The effect of early life exposure was much stronger than that of current pollution assessed by concentrations of microparticulates. Importantly, the findings raise the possibility that the health effects of air pollution have been underestimated as most studies do not have data on early life exposure, where vulnerability may be greatest, and even the effects of the lower levels of pollution experienced by current

Western populations appears to have been underappreciated. Indeed, a recent mortality study based on the Medicare population suggests that a significant disease burden is associated with pollutant levels that are below current statutory standards.[52] The evidence that air pollution has been having a major detrimental impact on health for decades and continues to do so underlines the importance of routine environmental tracking and surveillance systems in detecting and avoiding the harmful health effects of environmental pollution. The data presented in this analysis also have implications for the long-term health of the populations of countries that still depend on large amounts of coal for their domestic markets. This includes newly industrialised countries such as India or China, where coal is a major energy source, and resource-poor countries, where indoor air pollution from cook stoves results in heavy exposure of women and young children to pollution.[7]

**Acknowledgements** The authors would like to thank Laura Todd and her colleagues at the UK Office for National Statistics for their help in providing the mortality data. The authors would also like to thank David Coggon for helpful comments on the manuscript.

**Contributors** DIWP discovered the dataset on coal consumption in 1951/1952 and together with CO conceived the study. HS and PA advised on the use of the geographical data and carried out the redistricting analysis. DIWP, CO and AJ were responsible for the statistical analysis. The manuscript was written by DIWP, CO, AJ, HS and STH. All authors participated in a critical revision of the text and approved the final manuscript. DIWP is responsible for the overall content as the corresponding author.

**Funding** The work was supported by the Medical Research Council.

**Competing interests** None declared.

**Patient consent** Not required.

**Ethics approval** None

**Provenance and peer review** Not commissioned; externally peer reviewed.

**Data sharing statement** Under the MRC's 'Policy and Guidance on Sharing of Research Data from Population and Patient Studies', the dataset is available on request to diwp@mrc.soton.ac.uk

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
