## [Reviewer comments · BMJ Open]

ARTICLE DETAILS

TITLE (PROVISIONAL)	Evaluating the long-term consequences of air pollution in early life: geographical correlations between coal consumption in 1951/2 and current mortality in England and Wales.
AUTHORS	Phillips, David; Osmond, Clive; Southall, Humphrey; Aucott, Paula; Jones, Alexander; Holgate, Stephen

VERSION 1 – REVIEW

REVIEWER	Patrick Saunders carolan57 Ltd, UK
REVIEW RETURNED	30-Jun-2017

GENERAL COMMENTS	This is a timely research paper given the current high profile of air quality and health and a welcome addition to our understanding as to just how dangerous air pollution has been and continues to be. There is currently no known safe level of exposure to PM2.5 and a major open cohort study of over 60m people in the US (Qian et al Air Pollution and Mortality in the Medicare Population N Engl J Med 2017; 376:2513-2522) published yesterday has shown the burden of mortality associated with levels at or below current statutory standards-a reference to this study might be considered appropriate. The paper is well written and clearly acknowledges the limitations of the ecological design. The interpretation and conclusions are well defended and plausible. I think the authors have missed an opportunity to extol the virtue of a routine system of the ongoing routine tracking of environmental exposures and plausibly associated health outcomes. Two minor comments-although they can be derived from the methods I would have found definitions of infant and young children useful; the title doesn't strictly meet recommendation 1(a) of the STROBE checklist
--

REVIEWER	Jamie Mullins University of Massachusetts Amherst, United States
REVIEW RETURNED	28-Sep-2017

GENERAL COMMENTS	Summary: The article titled: Six decades after the Clean Air Act: Evaluating the health legacy of Coal seeks to link past exposure to air pollution from domestically (i.e.- household) burned coal to increased mortality rates. The study uses measures of different kinds of fuel usages by local government district in 1951/52 and finds that areas that had high domestic coal usage experience higher all-cause mortality rates from 1993 to 2012. The authors show that increases in mortality are focused in areas that are thought to be impacted by air pollution exposure such as cardiovascular, respiratory, and
---

certain cancer-linked deaths.

The paper addresses an important question regarding the long term effects of air pollution exposure, using an interesting data source and empirical strategy.

Comments:

Abstract: Results- It needs to be clear that coal “had the most powerful effects on mortality” of the factors considered. It would be nice to have a better idea of what the set of considered factors are even at this point in the abstract, otherwise the effects of coal should not be described relatively.

Strengths and limitations: first bullet point seems to ignore references 4, 18, & 19 among others.

Is the observation of analysis the 342 current LGDs for which mortality rates are available or the 1,145 LGDs for which fuel usage levels are known in 1951/52? The correct unit of analysis is the one for which real variation in the variables of interest exists in the data.

Two main issues with the analysis:

1.) The correlation between coal usage and population density means that the identified effects are really for the combined contribution of these two measures. This is true even if the contributions of domestic coal usage are “stronger” than those from population density.

2.) The effects of PM2.5 on mortality are estimated to be significant and beneficial in the main multi-variate analyses. If this cannot be explained (e.g.- higher pollution in Brittan today isn’t really that high, and the highest levels are associated with city centers where the wealthy live, and the wealthy tend to have longer than normal life expectancies), then we cannot take the other estimates from the analyses at face-value since there are clearly contributing factors which are important and left unaddressed by the current empirical approach.

The statistical approach used with the intent of separating the effects of population density and domestic coal use does not appear to be valid. The analysis presented seems (to this reader) to show that the variation in the variables that is the same is driving the effect while the variation that is different is not driving the main effect. In any case, a much better description of how and why the authors reach their conclusion to attribute all of the effects of these mechanically collinear variables onto air pollution via domestic coal usage is needed. If this is a standard approach, the source needs to be referenced.

Again, even if the authors are 100% justified in their conclusion that “the influence of domestic use was stronger than that of density”, the main results still attribute all the effects of density to domestic coal usage. The main estimates are thus only an upper bound on the effects claimed by the authors. If “stronger” means >50%, then the authors could provide a lower-bound for the effect of domestic coal use at 50% of their main estimates.

To be fair, the authors do not lean strongly on the exact magnitude of their estimates, but further qualifications are warranted in the discussion of results.

The authors frame domestic coal use as a measure of exposure during the period when the measure was taken, but isn’t domestic coal use in 1951/52 highly correlated with use at other times (acknowledged by authors) and air pollution generally at all times through the present? I’d like to see correlation measures of domestic coal use (as defined in this paper) with other measures of pollution (perhaps those that began in 1971 that were leveraged by reference #19). Perhaps the identified effects are lifetime exposure effects of higher pollution levels, which are being proxied-for by density and

	coal usage in 1951/52. The results are still interesting, but their interpretation changes. Along a different tack, if domestic coal burning is driving the effect, maybe it is exposure within the home to pollution that is driving the result. If this is the case, then perhaps Domestic Fuel Consumption per person or per household would be a better measure of exposure (and not collinear to population density). Maybe other confounders would start to bite for this analysis, but I'd appreciate at least a consideration of an alternative measure of domestic coal usage, especially if it might help tease out the mechanisms through which the effect is arising. Perhaps the significant effects on Tuberculosis are actually arising from the density not the coal. Are there other results that fit better with a population density explanation than a domestic coal use explanation? The creation of the social class adjustment variable should be clarified. I am aware of neither Brittan's "five social classes" nor how 5 population shares can be condensed into a single measure without imposing cardinal meaning onto a (seemingly) ordinal measures. How much higher class is class 4 than class 3 (or vis versa)? There is a significant literature looking at the health (and other) effects of the use of indoor cook stoves in developing countries. Is the domestic combustion of coal studied here similar to the settings examined by such studies? It seems similar enough to warrant citing and discussing the similarities of some such studies. Do different qualities of coal dominate in different regions? Are effects different in regions that had cleaner burning coal sources? Is there any better or alternative way to map mortality to exposure location? Specifically, could the mortality data be broken down by place of birth rather than place of death? Or better yet, place of residence in 1951/52? Are there other present day conditions that should be controlled for beyond PM2.5? Maybe social class, education, crowding, unemployment, industrial density at time of death (or in the death period) should be controlled for. Please address the possible confounding effects of such factors or convince the reader they are orthogonal to the studied relationship.
--	---

VERSION 1 – AUTHOR RESPONSE

Editorial requests:

- Please revise the title of your manuscript to include the research question, study design and setting. This is the preferred format of the journal.

We have changed the title to "Evaluating the early life effects of air pollution: geographical correlations between coal consumption in 1951/2 and current mortality in England and Wales.

Reviewer: 1

Reviewer Name: Patrick Saunders

Institution and Country: carolan57 Ltd, UK Please state any competing interests or state 'None declared': None declared

Please leave your comments for the authors below This is a timely research paper given the current high profile of air quality and health and a welcome addition to our understanding as to just how dangerous air pollution has been and continues to be. There is currently no known safe level of

exposure to PM2.5 and a major open cohort study of over 60m people in the US (Qian et al Air Pollution and Mortality in the Medicare Population N Engl J Med 2017; 376:2513-2522) published yesterday has shown the burden of mortality associated with levels at or below current statutory standards-a reference to this study might be considered appropriate. The paper is well written and clearly acknowledges the limitations of the ecological design. The interpretation and conclusions are well defended and plausible. I think the authors have missed an opportunity to extol the virtue of a routine system of the ongoing routine tracking of environmental exposures and plausibly associated health outcomes. Two minor comments-although they can be derived from the methods I would have found definitions of infant and young children useful; the title doesn't strictly meet recommendation 1(a) of the STROBE checklist

We thank Dr Saunders for his kind review of our work.

The recent NEJM paper is of great interest and we have commented on this and added a reference. (Page 13, line 8 up and reference 49).

We have also added a comment on the importance of routine environmental tracking and surveillance systems and their effects on health. (Page 13, line 6 up)

The title has been changed to "Evaluating the early life effects of air pollution: geographical correlations between coal consumption in 1951/2 and current mortality in England and Wales." which we hope is more explanatory.

The definitions of infant and young children do tend to be somewhat vague: in the methods section we have changed this to being in the first decade of life. (page 6, line 11 up).

Reviewer: 2

Reviewer Name: Jamie Mullins

Institution and Country: University of Massachusetts Amherst, United States Please state any competing interests or state 'None declared': None declared

Please leave your comments for the authors below

Summary:

The article titled: Six decades after the Clean Air Act: Evaluating the health legacy of Coal seeks to link past exposure to air pollution from domestically (i.e. - household) burned coal to increased mortality rates. The study uses measures of different kinds of fuel usages by local government district in 1951/52 and finds that areas that had high domestic coal usage experience higher all-cause mortality rates from 1993 to 2012. The authors show that increases in mortality are focused in areas that are thought to be impacted by air pollution exposure such as cardiovascular, respiratory, and certain cancer-linked deaths.

The paper addresses an important question regarding the long term effects of air pollution exposure, using an interesting data source and empirical strategy.

We thank Professor Mullins for reviewing our work and for his constructive comments.

Comments:

Abstract: Results- It needs to be clear that coal "had the most powerful effects on mortality" of the factors considered. It would be nice to have a better idea of what the set of considered factors are even at this point in the abstract, otherwise the effects of coal should not be described relatively.

We have reworded the abstract (results section) which now reads “Of all the sources of solid fuel, domestic usage had the strongest effect on mortality”.

Strengths and limitations: first bullet point seems to ignore references 4, 18, & 19 among others.

We have emphasized that our study is a mortality study but still stand by our assertion that this is the first long term study of the relationship between air pollution and late life mortality. We are not aware of a similar study and the cited references above do not provide data over such a long period.

Is the observation of analysis the 342 current LGDs for which mortality rates are available or the 1,145 LGDs for which fuel usage levels are known in 1951/52? The correct unit of analysis is the one for which real variation in the variables of interest exists in the data.

The abstract has been revised to indicate that the unit of analysis was the current 342 LGDs (only in this form is the current mortality data available). The methods section describes how the fuel usage data, which were published for the 1,145 LGDs in 1951/2 were amalgamated to accord with the current LGDs. (Page 5, second para)

Two main issues with the analysis:

1.) The correlation between coal usage and population density means that the identified effects are really for the combined contribution of these two measures. This is true even if the contributions of domestic coal usage are “stronger” than those from population density.

Please see response below.

2.) The effects of PM2.5 on mortality are estimated to be significant and beneficial in the main multi-variate analyses. If this cannot be explained (e.g.- higher pollution in Brittan today isn't really that high, and the highest levels are associated with city centers where the wealthy live, and the wealthy tend to have longer than normal life expectancies), then we cannot take the other estimates from the analyses at face-value since there are clearly contributing factors which are important and left unaddressed by the current empirical approach.

We thank you for pointing this out. Current pollution levels are largely derived from traffic pollution and, as you say, tend to be highest in city centres although there is likely to be heterogeneity in the wealth profile of those most affected. The univariate effects of the PM2.5 data are consistent with the published effects of this dataset on current mortality (reference 19) The changed direction of the regression coefficient in the multivariate analyses is likely to have arisen because of the well-known effect of correlation between our measure of domestic air pollution and current PM2.5 levels resulting in suppression of the effect of current PM2.5. However, it must be emphasized that our analysis is structured to examine possible confounders of the relationship between coal pollution and current mortality rather than to evaluate the effects of current pollutants which has been done extensively by others.

The statistical approach used with the intent of separating the effects of population density and domestic coal use does not appear to be valid. The analysis presented seems (to this reader) to show that the variation in the variables that is the same is driving the effect while the variation that is different is not driving the main effect. In any case, a much better description of how and why the authors reach their conclusion to attribute all of the effects of these mechanically collinear variables onto air pollution via domestic coal usage is needed. If this is a standard approach, the source needs to be referenced.

Again, even if the authors are 100% justified in their conclusion that “the influence of domestic use was stronger than that of density”, the main results still attribute all the effects of density to domestic coal usage. The main estimates are thus only an upper bound on the effects claimed by the authors. If “stronger” means >50%, then the authors could provide a lower-bound for the effect of domestic coal use at 50% of their main estimates.

To be fair, the authors do not lean strongly on the exact magnitude of their estimates, but further qualifications are warranted in the discussion of results.

The approach used to separate the effects of population density and domestic coal usage has been used by others and we have added a reference (45). There is also an analogy with the commonly used “Bland - Altman” plot, which evaluates differences in measurement methods by separating and displaying the mean and difference of their values.

The sum and difference approach tells us the following. The sum of the effects of domestic usage and density gives us an estimate of the combined effect of both variables, which is strong (a 5.4% increase in overall death rate per SD change in these predictors using the adjusted data). The difference regression gives us an estimate of the extent to which the effect of coal consumption is stronger than that of population density (3.1% increase in overall death rate per SD) but they do not enable us to estimate the actual effects of each variable in a regression model.

It is also worth pointing out that indices of density are probably less important than overcrowding in determining health outcomes. Although the work done on this has lacked rigour, overcrowding and particularly childhood overcrowding seems to be a more consistent predictor of health outcomes than density and all the regression models in table 2 suggest that the effect of coal consumption is independent of overcrowding.

However, we agree that we need to be more cautious in interpreting the findings with respect to the density data and have altered the discussion to indicate this qualification. (page 13, first para)

The authors frame domestic coal use as a measure of exposure during the period when the measure was taken, but isn't domestic coal use in 1951/52 highly correlated with use at other times (acknowledged by authors) and air pollution generally at all times through the present? I'd like to see correlation measures of domestic coal use (as defined in this paper) with other measures of pollution (perhaps those that began in 1971 that were leveraged by reference #19). Perhaps the identified effects are lifetime exposure effects of higher pollution levels, which are being proxied-for by density and coal usage in 1951/52. The results are still interesting, but their interpretation changes.

Tables for coal consumption in the UK, particularly that used domestically, are available online (reference 11) and they show that consumption was high in the post-war period and remained high during the 1950s falling thereafter because of the switch to other fuels and the impact of the clean air act. Representative figures (million tons p.a.) are 1952: 37, 1960: 36 1970:20 1980:9, and then falling rapidly This gives us a natural experiment whereby we can look at the UK mortality among people who were exposed in early life but not in later life, excepting motor vehicle pollution.

Providing data on the correlation between measures of coal use at this time and actual pollution is a problem for several reasons. Firstly, the routine monitoring system didn't get going until several years after this dataset and the 1971 data you refer to were measured 20 years later when, as you see above, there was a considerable reduction in coal usage. Secondly, the early measurements used crude technology and didn't measure particulates as we do now. Thirdly, the data were from a relatively few isolated monitoring stations and did not capture the very large geographical variations in an area and sometimes failed to capture diurnal and seasonal variation. In the absence of such

measurements we argue that the consumption data we have over a year represent a reasonable integral of pollution exposure in a given locality.

Along a different tack, if domestic coal burning is driving the effect, maybe it is exposure within the home to pollution that is driving the result. If this is the case, then perhaps Domestic Fuel Consumption per person or per household would be a better measure of exposure (and not collinear to population density). Maybe other confounders would start to bite for this analysis, but I'd appreciate at least a consideration of an alternative measure of domestic coal usage, especially if it might help tease out the mechanisms through which the effect is arising.

We thank you for this suggestion and indeed have carried out analyses based on domestic fuel consumption per household or per person finding that the pattern of associations is similar although with rather reduced risk estimates. The effects, as you suggest, were independent of density. While these measures do reduce collinearity with population density, we consider that they would be a poorer measure of population exposure because the amount of pollution derives predominantly from the total quantity of coal burnt in an area and the resulting poor air quality and smog rather than just pollution within the household, although this is likely to have been a factor. Most British houses in the post-war era had chimneys and, although these were of variable efficiency, most of the combustion products would have been released into the local environment.

Perhaps the significant effects on Tuberculosis are actually arising from the density not the coal. Are there other results that fit better with a population density explanation than a domestic coal use explanation?

While we accept this criticism, the literature is more supportive of a relationship between overcrowding and TB than with density (reviewed in reference 46) and our data suggest that the relationship between pollution and TB appears to be independent of overcrowding both in early life and adulthood (Table 4). However, we have modified the text to admit this possibility. (Page 11, line 10 up)

The creation of the social class adjustment variable should be clarified. I am aware of neither Brittan's "five social classes" nor how 5 population shares can be condensed into a single measure without imposing cardinal meaning onto a (seemingly) ordinal measures. How much higher class is class 4 than class 3 (or vis versa)?

The Registrar General's system of classifying individuals into social class groups has been in use in Britain for most of the 20th century. It is an occupation-based system which categorises people into five ordinal groups, which consistently predict mortality and lifespan (see Donkin, Goldblatt and Lynch: *Health Statistics Quarterly*, 2002:15:5-15). The groups are (I) Professional occupations, (II) Managerial and Technical occupations, (III) Skilled occupations, (IV) Partly-skilled occupations and (V) Unskilled occupations. Because of their ordinal nature, it is standard practice to combine these into a single measure. Indeed, repeating the regression analyses using the proportions in each social class as dummy variables generates identical regression coefficients.

In response, we have modified the methods section to describe better the social class variables used (page 6, first para) and have added a reference (ref 18).

There is a significant literature looking at the health (and other) effects of the use of indoor cook stoves in developing countries. Is the domestic combustion of coal studied here similar to the settings examined by such studies? It seems similar enough to warrant citing and discussing the similarities of some such studies.

This is a good point and indeed was the motivation for the lead author (DIWP) in doing these studies. The question of similarities between UK air pollution and pollution in resource-poor countries is, however, hard to answer as a wide variety of different fuels are used in developing countries and much of the exposure is within the home because of the poor smoke venting arrangements. However, the important point from the present study is that the effects of smoke are not merely related to respiratory disease but are associated with a wide variety of other diseases, which continue to be of major importance in developing countries, such as rheumatic heart disease, tuberculosis and even hypertension, which is very prevalent in rural areas of Africa. It also underlines the impact of childhood exposure as being particularly important for the long term effects.

We have therefore added a referenced section to discuss importance of these findings in relationship to the problems of domestic air pollution in developing countries. (Pages 4, para 1 and 13 last para and page 14)

Do different qualities of coal dominate in different regions? Are effects different in regions that had cleaner burning coal sources?

There was certainly variation in the proportions of smokeless fuels used in different areas but this didn't show a clear geographical pattern. For examples, while wealthier areas tended to use more smokeless fuels, these were also used in certain mining communities where they were more available. We looked at smokeless fuels in the analysis but found, as one might expect, that they contributed little to the overall mortality risks so we used non-smokeless domestic fuel in the final analyses.

Is there any better or alternative way to map mortality to exposure location? Specifically, could the mortality data be broken down by place of birth rather than place of death? Or better yet, place of residence in 1951/52?

We agree that this would be a really helpful analysis to do. However, place of birth was not routinely recorded on death certificates in the UK with the exception of a trial period between 1969 and 1972. Are there other present day conditions that should be controlled for beyond PM2.5? Maybe social class, education, crowding, unemployment, industrial density at time of death (or in the death period) should be controlled for. Please address the possible confounding effects of such factors or convince the reader they are orthogonal to the studied relationship.

Thank you for this suggestion and we agree that these should be included. We have therefore reworked the analysis adjusting for the current socioeconomic influences (as above) and for current PM 2.5 at the time of death and present revised tables in the analysis. These are summarized in the new Table 3 and the disease-specific effects in Figure 2 and Tables 4 & 5, show crude effects and effects adjusted for the same variables. The impact of these adjustments is small and this is now described in the results section and considered in the discussion (pages 8, first para and 12, line 6 up).

VERSION 2 – REVIEW

REVIEWER	patrick saunders University of Staffordshire, UK
REVIEW RETURNED	11-Nov-2017
GENERAL COMMENTS	The authors have addressed my comments in this revision but I have to defer to the 2nd reviewers comments on the statistical analysis. My recommendation to accept is accordingly dependent on

	the 2nd reviewers recommendation
REVIEWER	Jamie Mullins University of Massachusetts Amherst
REVIEW RETURNED	22-Nov-2017

GENERAL COMMENTS	I am grateful for the careful consideration paid by the authors to my previous comments and questions. The responses to my questions and changes made to the paper are most appreciated. I believe the paper is stronger now that at the time of original submission. With that being said, the abstract and introduction need some additional attention. While the important substantive content is included in these sections, both warrant some polishing to improve flow and clarify the exposition. Below are a number of specific comments that I believe would improve the article further. Comments: Although the authors caveat the uniqueness claims in the body of the paper, the claim in the “Objective” section of the abstract still rings hollow. Here is a review article looking at the long-term effects of early life exposure to pollution: Currie, Janet, et al. "What do we know about short-and long-term effects of early-life exposure to pollution?." Annu. Rev. Resour. Econ. 6.1 (2014): 217-247. Granted, this review doesn't focus on medical journal articles, but the methods employed by the authors are more akin to the analysis covered in the Currie article than to traditional approaches in the medical literature. In the “Main outcome measure” portion of the abstract, the meaning of the term “geographically equivalent” is unclear. In the “Results” section of the abstract, “domestic usage” seems to be used to describe “sources of solid fuel”, which I don't think is correct. In the “Conclusions” section of the abstract, the word “Although” at the start of the 2nd sentence appears unnecessary. The same sentence states that the “data raise the possibility...”. It seems than the analysis or results “suggest” would be a more accurate characterization of the purpose of the paper. The first paragraph of the text is helpful in framing the importance of the results, but should be worked over again to improve flow and clarity. The authors currently cite Logan (1952) for the mortality estimate of the Great Smog (at ~4,000), but today it is more common to reference ~12,000 deaths (taking into account higher death rates in the months after the event) as estimated by: Bell, Michelle L., Devra L. Davis, and Tony Fletcher. "A retrospective assessment of mortality from the London smog episode of 1952: the role of influenza and pollution." Environmental health perspectives 112.1 (2004): 6. The 3rd paragraph of the body of the text references “such vast quantities” before any quantities are referenced. In the first paragraph of the methods section the term “Domestic supplies” is used. The meaning of this term, or the concept meant to be addressed, needs to be clarified. Same paragraph, the reference to smokeless fuel should include a description of what fuels fall into this category and a note that the effects of these fuels are considered in the analysis. References to tables and figures should have the words capitalized
---

	(or not) consistently. P. 8- Again, the discussion of smokeless fuels seems awkwardly placed and incomplete. P. 10- the authors note: "The consumption measures in our study were shown to correlate with subsequent measurements of pollution in the UK." This seems to suggest that coal usage (the authors' proxy for pollution in 1951-52) may actually be a proxy for the level of pollution in a given area over the long-term. The identified effects (or correlations) would therefore be better characterized as the effects of long-term exposure rather than the long-term effects of past exposure. P.12- related to the last point, the authors state that a confounding variable that would be problematic for their analysis would "have to correlate very strongly with both domestic air pollution and respiratory disease". First off, "domestic air pollution" seems to suggest a different quantity than the authors are seeking to communicate. Second, it would seem that "air pollution over the long-term" might be such a variable. As I noted in my prior review, this doesn't invalidate the importance of the paper or the analysis, it just suggests a slightly different framing of the results, or at least an acknowledgement. P.13- The discussion of density vs. overcrowding is confusing because overcrowding is not defined. I think the point is that density matters, but in a non-linear way. Since the authors identify a linear effect of their pollution measure, their results cannot be driven by population density. This is a convincing argument and should be clarified. Figure 2 could be referenced again to demonstrate the lack of a threshold or other non-linear effect. Table 3- It seems odd that mortality risk is increasing in Social Class and Education, no? Unless there is a compelling reason to show the univariate estimates for all the SES controls (such as to demonstrate that they are related to the outcome variable in sensible ways) it doesn't seem necessary to include these estimates.
--	--

VERSION 2 – AUTHOR RESPONSE

Reviewer: 2

Please leave your comments for the authors below I am grateful for the careful consideration paid by the authors to my previous comments and questions. The responses to my questions and changes made to the paper are most appreciated. I believe the paper is stronger now than at the time of original submission.

With that being said, the abstract and introduction need some additional attention. While the important substantive content is included in these sections, both warrant some polishing to improve flow and clarify the exposition. Below are a number of specific comments that I believe would improve the article further.

We thank you for these helpful comments and suggestions on our manuscript

Comments:

Although the authors caveat the uniqueness claims in the body of the paper, the claim in the "Objective" section of the abstract still rings hollow. Here is a review article looking at the long-term effects of early life exposure to pollution:

Currie, Janet, et al. "What do we know about short-and long-term effects of early-life exposure to pollution?." *Annu. Rev. Resour. Econ.* 6.1 (2014): 217-247.

Granted, this review doesn't focus on medical journal articles, but the methods employed by the authors are more akin to the analysis covered in the Currie article than to traditional approaches in the medical literature.

We have changed the wording of the objective section of the abstract. (Abstract) We also thank you for alerting us to the Currie review which is now cited in the introduction section.(ref 6)

In the "Main outcome measure" portion of the abstract, the meaning of the term "geographically equivalent" is unclear.

This has been omitted and the sentence reworded (abstract; main outcome measures)

In the "Results" section of the abstract, "domestic usage" seems to be used to describe "sources of solid fuel", which I don't think is correct.

This has been changed to "Domestic (household) coal consumption" (Abstract; results section)

In the "Conclusions" section of the abstract, the word "Although" at the start of the 2nd sentence appears unnecessary. The same sentence states that the "data raise the possibility...". It seems than the analysis or results "suggest" would be a more accurate characterization of the purpose of the paper.

These changes have been incorporated. (Abstract: Conclusion section)

The first paragraph of the text is helpful in framing the importance of the results, but should be worked over again to improve flow and clarity.

This has been reworked to make it clearer and a note added on the economic impacts of pollution.(page 4: para 1)

The authors currently cite Logan (1952) for the mortality estimate of the Great Smog (at ~4,000), but today it is more common to reference ~12,000 deaths (taking into account higher death rates in the months after the event) as estimated by:

Bell, Michelle L., Devra L. Davis, and Tony Fletcher. "A retrospective assessment of mortality from the London smog episode of 1952: the role of influenza and pollution." *Environmental health perspectives* 112.1 (2004): 6.

This has been changed and the new reference cited. (page 4, line 17)

The 3rd paragraph of the body of the text references "such vast quantities" before any quantities are referenced.

This has been reworded. (Page 4, lines 8-9 up)

In the first paragraph of the methods section the term "Domestic supplies" is used. The meaning of this term, or the concept meant to be addressed, needs to be clarified.

This has been reworded to make it clearer.(page 5 para 1)

Same paragraph, the reference to smokeless fuel should include a description of what fuels fall into this category and a note that the effects of these fuels are considered in the analysis.

The fuels included in the smokeless category are now listed (page 5, first para) and a note added to the effect that there was a separate analysis of their impact. (page 7, line 13 up).

References to tables and figures should have the words capitalized (or not) consistently.

These have been corrected.

P. 8- Again, the discussion of smokeless fuels seems awkwardly placed and incomplete.

See above: this section has now been moved.

P. 10- the authors note: "The consumption measures in our study were shown to correlate with subsequent measurements of pollution in the UK." This seems to suggest that coal usage (the authors' proxy for pollution in 1951-52) may actually be a proxy for the level of pollution in a given area over the long-term. The identified effects (or correlations) would therefore be better characterized as the effects of long-term exposure rather than the long-term effects of past exposure.

The data correlating consumption with actual pollution measures were obtained in the early 1960s when coal was still an important source of pollution. We point out that although pollution in the years adjacent to 1951/2 are likely to be similar, the impact of the clean air act (which created clean air zones) and changing fuel usage led to a steep drop in coal consumption during and after the 1960s and dramatic falls in resulting pollution. So while we accept that this is not a sharply defined past exposure, it still represents a way of assessing early life impacts of pollution.

In response we have altered the last paragraph on page 9 (highlighted section) and mention that we have not got data on all possible pollutants although these are likely to be of minor importance.

P.12- related to the last point, the authors state that a confounding variable that would be problematic for their analysis would "have to correlate very strongly with both domestic air pollution and respiratory disease". First off, "domestic air pollution" seems to suggest a different quantity than the authors are seeking to communicate. Second, it would seem that "air pollution over the long-term" might be such a variable. As I noted in my prior review, this doesn't invalidate the importance of the paper or the analysis, it just suggests a slightly different framing of the results, or at least an acknowledgement.

We take this point and have added a sentence to this effect (page 11, line 4 up)

P.13- The discussion of density vs. overcrowding is confusing because overcrowding is not defined. I think the point is that density matters, but in a non-linear way. Since the authors identify a linear effect of their pollution measure, their results cannot be driven by population density. This is a convincing argument and should be clarified. Figure 2 could be referenced again to demonstrate the lack of a threshold or other non-linear effect.

This is a helpful point and we have included a sentence raising this in the discussion (page 13, line 18 down) and defined overcrowding (page 13, line 12 down)

Table 3- It seems odd that mortality risk is increasing in Social Class and Education, no? Unless there is a compelling reason to show the univariate estimates for all the SES controls (such as to demonstrate that they are related to the outcome variable in sensible ways) it doesn't seem necessary to include these estimates.

The reason for this is that our social class score increases with lower social class (a standard way of doing this analysis) while education refers to the proportion with no qualifications. The tables have been modified to make this clear (Tables 2 and 3). We prefer to keep the univariate estimates of SES factors in the analysis as they demonstrate effect size and direction of effect.

VERSION 3 – REVIEW

REVIEWER	L. Pokhrel Temple University, USA
REVIEW RETURNED	17-Feb-2018

GENERAL COMMENTS	The paper by Phillips et al. attempts to evaluate the impact of early life air pollution on subsequent mortality after 60 years. Using ecological method and limited coal-use data, the study goes on to suggest that coal-based pollution, experienced over 60 years ago in early life, affects human health after 60 years by increasing mortality from a wide variety of diseases. However, the study has many limitations—some I outline below—and thus I find it unsuitable for publication in MBJ Open. The study did not include information about exposure to other pollutants; many hazardous pollutants, also called 'Air Toxics', are also being released from various other sources other than coal/fuel use and people are being exposed to on a regular basis, albeit at different levels depending on the area of residence, occupation, etc. Smoking is just another factor that can manifest similar health outcomes but was not considered in this study, hence severely impacting the study conclusions. The study did not also account for migration issue as both in-migration and out-migration could introduce strong confounding effect on the relationship sought. Only one-two years of coal use data are used to show correlation with later-life health outcomes. While there is consensus in toxicology that early life exposures are critical in one's development, it cannot be solely responsible for later-life outcomes as we as humans are also exposed to various toxicants at various levels throughout our life and for the onset of various chronic illnesses as studied in this paper exposures must occur for many years over the lifetime. Further, those individuals who were 35-40 years of age in 1993 were not even born during 1951-1952, the time for which coal-use data were available and used in this study. Hence, using 1951-52 coal use data to predict health outcomes after 60 years using simple measure of statistics is a bit far-fetched, if not outright strange. More importantly, tuberculosis was the one that showed strongest association, while, in fact, tuberculosis is caused by a bacterium Mycobacterium tuberculosis, NOT by pollution due to coal combustion. Showing correlation between coal and tuberculosis is nothing but wrong and show utter lack of knowledge of disease
--

	manifestation. Overall, because this is a geographical observational study, a mere correlation as shown in this study MUST not be misconstrued as cause-effect relation. Given the significant uncertainties in the study findings and conclusions drawn, I would not recommend publication of this manuscript in BMJ Open.
--	--

REVIEWER	Joshua Warren Department of Biostatistics, Yale University; USA.
REVIEW RETURNED	26-Feb-2018

GENERAL COMMENTS	Statistical modeling questions: [1] The authors do not appear to account for spatial correlation in the observed data through use of a spatial random effect (or other method) in the Poisson regression framework. Failure to account for spatial correlation in a geographic analysis can lead to confidence intervals that are narrower than they should be. This could potentially lead to the identification of statistically significant associations that are incorrect. [A] Can the authors provide evidence to suggest that spatial correlation was not an important feature in these data (residual analysis using Moran's I perhaps)? [B] Or can the authors reanalyze the data while including spatially referenced random effect for areal data (conditional autoregressive model)? [2] Can the authors provide a reference for the Fisher-Yates transformation? [3] There are a number of results displayed/discussed from separately fit models. Did the authors consider some type of correction for multiple comparisons/testing (maybe Bonferroni)? How many of these findings remain statistically significant after a correction?
---

VERSION 3 – AUTHOR RESPONSE

Reviewer: 3

Reviewer Name

L. Pokhrel

Institution and Country

Temple University, USA

Please state any competing interests or state 'None declared':

None

Please leave your comments for the authors below The paper by Phillips et al. attempts to evaluate the impact of early life air pollution on subsequent mortality after 60 years. Using ecological method and limited coal-use data, the study goes on to suggest that coal-based pollution, experienced over 60 years ago in early life, affects human health after 60 years by increasing mortality from a wide variety of diseases. However, the study has many limitations—some I outline below—and thus I find it unsuitable for publication in MBJ Open.

The study did not include information about exposure to other pollutants; many hazardous pollutants, also called 'Air Toxics', are also being released from various other sources other than coal/fuel use and people are being exposed to on a regular basis, albeit at different levels depending on the area of residence, occupation, etc.

We were unable to include information on other air pollutants "Air Toxics" from sources other than the combustion of coal and other solid fuels as these data are unavailable for the early 1950s. However, our sensitivity analyses (page 11, last line) suggest that there would have to be a very high correlation between unknown pollutants and both fuel use and subsequent mortality to confound the relationships observed. As many Air Toxics of current concern would have been geographically localised and specific to certain industries it is difficult to see how they could be confounders especially as our analyses incorporate adjustment for socioeconomic factors (eg occupation/urbanisation) that would reflect differential exposure to these substances. In response to this we have included a section discussing this issue. (page 12, first para).

Smoking is just another factor that can manifest similar health outcomes but was not considered in this study, hence severely impacting the study conclusions.

Regional smoking data did not become available in the UK until much later and so could not be included in this analysis. We do not think that this seriously confounded the study as in the early 1950s smoking was a social norm with very high rates (78% of men used tobacco in 1951, Cancer Research, UK). Furthermore, smoking is associated with socioeconomic status, which was included in our analyses and, finally, standardisation of the data for lung cancer rates, as a proxy for smoking, did not explain the correlations we observed. We include a section discussing the impact of smoking (page 12, last para).

The study did not also account for migration issue as both in-migration and out-migration could introduce strong confounding effect on the relationship sought.

We agree that migration is a potential problem with our study and include a substantial section discussing this. (page 11 line 22).

Only one-two years of coal use data are used to show correlation with later-life health outcomes. While there is consensus in toxicology that early life exposures are critical in one's development, it cannot be solely responsible for later-life outcomes as we as humans are also exposed to various toxicants at various levels throughout our life and for the onset of various chronic illnesses as studied in this paper exposures must occur for many years over the lifetime.

We do not state that early life exposures are "solely responsible" for later-life outcomes rather that early pollution "increases mortality" from a variety of diseases (page 13, line 5). Clearly other factors contribute to the various chronic diseases referred to in the paper in addition to the effects of early pollution. We would also point out that our analysis includes data on later life microparticulate air pollution.

Further, those individuals who were 35-40 years of age in 1993 were not even born during 1951-1952, the time for which coal-use data were available and used in this study.

The reason the individuals in this age range (which constituted 2.2% of the total deaths) were retained was to carry out the cohort analysis (figure 3) which explores the influence of birth year to subsequent mortality.

Hence, using 1951-52 coal use data to predict health outcomes after 60 years using simple measure of statistics is a bit far-fetched, if not outright strange. More importantly, tuberculosis was the one that showed strongest association, while, in fact, tuberculosis is caused by a bacterium *Mycobacterium tuberculosis*, NOT by pollution due to coal combustion. Showing correlation between coal and tuberculosis is nothing but wrong and show utter lack of knowledge of disease manifestation.

The reason for our study is the lack of information as to the long term effects of early life exposure to pollution in terms of hard outcomes such as mortality from major diseases. Ecological studies such as the present one are used in epidemiology to generate hypotheses rather than for the purpose of hypothesis testing and indeed it was our purpose to explore the range of potential effects of early air pollution from fossil fuels.

The history of air pollution in the UK gives us an opportunity to test this - it can be considered a natural experiment as high levels of exposure in the early 1950s declined rapidly after the clean air act. Because air pollution contains a wide variety of cellular toxins that impact on a variety of developing organs and systems, one would expect a range of diseases to be affected and it is indeed of interest to see the range of conditions which are linked and, conversely, those which are not.

The reviewer takes us to task for looking at correlations between pollution and tuberculosis. We agree that *M. tuberculosis* is the organism responsible for the disease but would point out that susceptibility to infection and disease progression in infected individuals is determined largely by the response of the host immune system. This is affected by intrinsic factors (such as immune system genetics) as well as extrinsic factors such as insults to the immune system which include pollution. Indeed, others

have established a role for both indoor and outdoor pollution and increased risk of developing pulmonary TB

See, for example, Narasimhan P, Wood J, Macintyre CR, Mathai D. *Pulm Med.* 2013;2013:828939.

Overall, because this is a geographical observational study, a mere correlation as shown in this study MUST not be misconstrued as cause-effect relation.

We do not claim that our results show a cause-effect relationship. However, we have changed some of the wording in the abstract to indicate that we are reporting associations. (Abstract: objective and results sections).

Given the significant uncertainties in the study findings and conclusions drawn, I would not recommend publication of this manuscript in *BMJ Open*.

Reviewer: 4

Reviewer Name

Joshua Warren

Institution and Country

Department of Biostatistics, Yale University; USA.

Please state any competing interests or state 'None declared':

None declared.

Please leave your comments for the authors below Statistical modeling questions:

[1] The authors do not appear to account for spatial correlation in the observed data through use of a spatial random effect (or other method) in the Poisson regression framework. Failure to account for spatial correlation in a geographic analysis can lead to confidence intervals that are narrower than they should be. This could potentially lead to the identification of statistically significant associations that are incorrect.

[A] Can the authors provide evidence to suggest that spatial correlation was not an important feature in these data (residual analysis using Moran's I perhaps)?

[B] Or can the authors reanalyze the data while including spatially referenced random effect for areal data (conditional autoregressive model)?

We have carried out an analysis to evaluate the effect of spatial autocorrelation on our data. These indicate relatively small effects which do not materially change the conclusions. These are now reported in a section included in the discussion (Page 11 line 13) and described in detail in a second appendix (appendix 2).

[2] Can the authors provide a reference for the Fisher-Yates transformation?

A reference is now provided (reference 22)

[3] There are a number of results displayed/discussed from separately fit models. Did the authors consider some type of correction for multiple comparisons/testing (maybe Bonferroni)? How many of these findings remain statistically significant after a correction?

Because the major considerations in this study are confounding rather than significance testing and as estimates of variability rather than p-values are being reported, we did not carry out Bonferroni corrections. We did consider adjusting confidence intervals to take account of the number of results displayed but because the study is based on a very large numbers of subjects with very small confidence intervals, this did not affect the conclusions.